# From Replication to Redesign: Exploring Pairwise Comparisons for LLM-Based Peer Review

Yaohui Zhang[1]    Haijing Zhang[1]    Wenlong Ji[1]
Tianyu Hua[1]    Nick Haber[1]    Hancheng Cao[2]    Weixin Liang[1]
[1] Stanford University    [2]Emory University

## Abstract

The advent of large language models (LLMs) offers unprecedented opportunities to reimagine peer review beyond the constraints of traditional workflows. Despite these opportunities, prior efforts have largely focused on replicating traditional review workflows with LLMs serving as direct substitutes for human reviewers, while limited attention has been given to exploring new paradigms that fundamentally rethink how LLMs can participate in the academic review process. In this paper, we introduce and explore a novel mechanism that employs LLM agents to perform pairwise comparisons among manuscripts instead of individual scoring. By aggregating outcomes from substantial pairwise evaluations, this approach enables a more accurate and robust measure of relative manuscript quality. Our experiments demonstrate that this comparative approach significantly outperforms traditional rating-based methods in identifying high-impact papers. However, our analysis also reveals emergent biases in the selection process, notably a reduced novelty in research topics and an increased institutional imbalance. These findings highlight both the transformative potential of rethinking peer review with LLMs and critical challenges that future systems must address to ensure equity and diversity.

## 1    Introduction

The peer review system serves as a cornerstone for validating and disseminating scientific ideas [2]. Yet, despite its widespread adoption, the peer review system has become increasingly strained because of the rapid growth of research output and a persistent shortage of qualified reviewers [13, 30, 40].

Recent advances in large language models (LLMs) [1, 16, 37] have sparked growing interest in using AI to assist or even automate parts of the review process [17, 22, 39]. Unlike human reviewers, whose availability and productivity are inherently constrained, LLM-based agents offer the potential for greater scalability and consistency in academic evaluation.

However, existing LLM-based paper review efforts [18, 27, 49, 55, 61] largely mirror traditional workflows: For each paper, multiple LLM agents first independently generate structured reviews (e.g., summary, strengths, weaknesses, suggestions for improvement, numeric ratings). Subsequently, an LLM meta-reviewer synthesizes these individual LLM reviews and composes a cohesive meta-review summarizing the collective assessment. Finally, a decision (e.g., accept or reject) is generated either directly by this LLM meta-reviewer agent or based on its synthesized ratings.

While LLMs offer new possibilities for peer review, simply replicating existing workflows may overlook a deeper opportunity. Historical analysis suggests that the conventional structure of peer review was not carefully engineered for optimal evaluation, but rather emerged as a pragmatic response to resource constraints. Studying the development of peer review in sociology, Merriman [32] observes that "the evolution of peer review is best understood as the product of continuous efforts to steward editors' scarce attention while preserving an open submission policy that favors authors' interests."

39th Conference on Neural Information Processing Systems (NeurIPS 2025).

This perspective highlights that many features of today's review systems reflect historical compromises rather than principled design choices. As scholarly publishing expands globally and across disciplines [10], these inherited structures have come under increasing scrutiny. Issues like bias [20, 44, 46] and noisy ratings [28] have spurred debates on how to improve or reform peer review [14, 45, 48, 52].

Given their exceptional scalability, LLMs present an opportunity not merely to automate existing practices, but to fundamentally rethink the design of manuscript evaluation systems. Rather than replicating workflows optimized for human reviewers, it is crucial to rethink the foundations of effective evaluation and to design AI systems that complement and address the persistent limitations of traditional peer review.

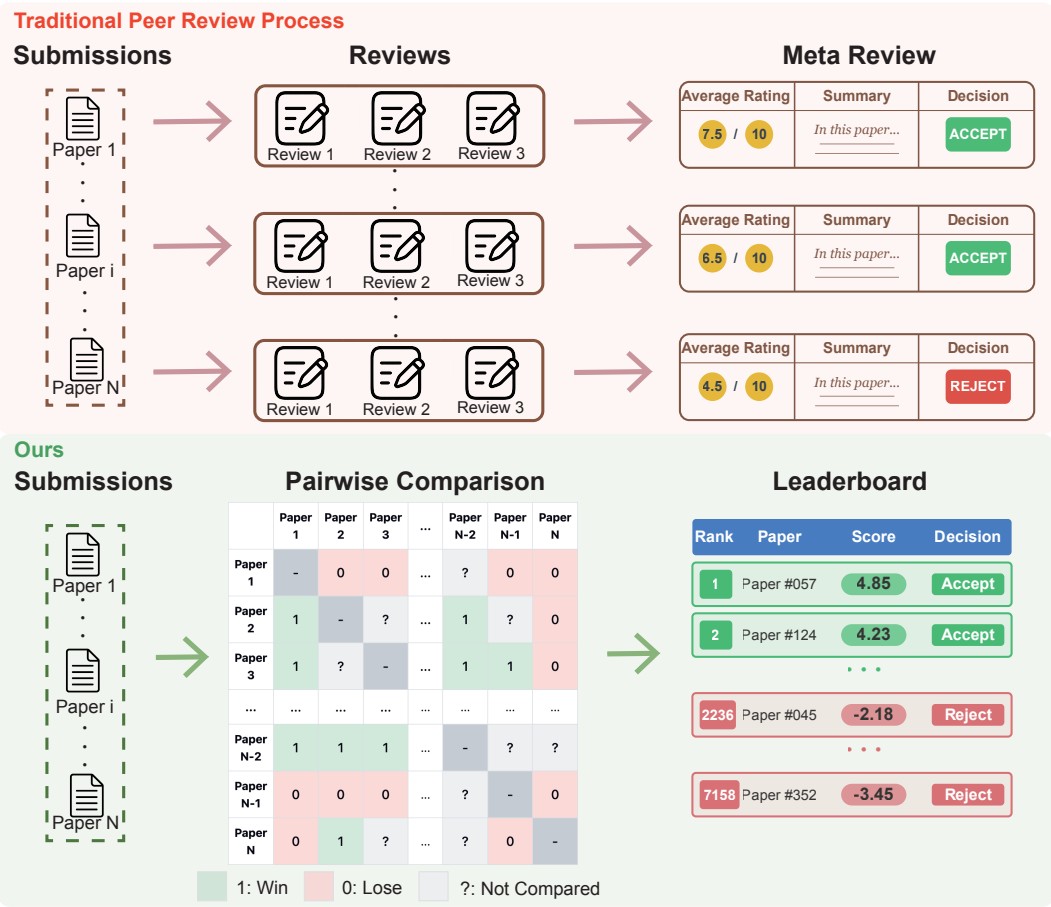

Figure 1: **Comparison of the conventional peer review process (top) and the proposed pairwise evaluation ranking system.** The traditional paradigm (followed by both human reviewers and existing LLM-based review systems) works on paper-level assessment, where each manuscript is first evaluated independently by multiple reviewers who provide detailed comments and scores. These evaluations are integrated by a meta-reviewer to recommend a final decision. Our method adopts pairwise comparisons between papers: each agent is randomly assigned a pair of submissions to evaluate. These comparative judgments are then aggregated using the Bradley–Terry model, where the resulting BT coefficients serve as scoring functions to produce a global ranking of all submissions. Final decisions are derived from the aggregated rankings, enabling a more objective assessment of the relative quality across the entire submission pool.

In this work, we propose a shift from replication to redesign. We focus on the **decision-making layer** of peer review—specifically, mechanisms that can rank submissions and support acceptance decisions. Our approach is intended to complement, not replace, the current peer review process. Specifically, we introduced and explored a novel mechanism that engages LLM agents in pairwise comparisons for more effective manuscript assessment. Rather than assigning absolute scores to

individual papers, our approach directly contrasts pairs of submissions to determine which one better meets the multidimensional evaluation criteria. Each LLM agent is tasked with evaluating two manuscripts at a time, and by integrating the results from numerous pairwise comparisons, we can construct a robust overall assessment that more accurately reflects relative quality.

Our results indicate that with proper scaling, this pairwise mechanism demonstrates potential to identify papers of higher academic impact (as measured by future citation count), significantly outperforming paper-level rating approach. However, our analysis also reveals important limitations: Papers preferred by our LLM-based pairwise ranking system tend to exhibit lower topic novelty and greater concentration among a limited number of high-prestige institutions. While these biases highlight critical challenges for the deployment of LLM-based review systems, we view this work as a first step toward exploring fundamentally new paradigms for scalable, equitable, and robust research evaluation.

It is important to note that while we use proxy metrics—such as future citation counts—to evaluate our proposed mechanism, this represents an exploratory and pragmatic step rather than an ideal. The gold standard would be direct human evaluation of how well a mechanism supports the selection and improvement of high-quality research, but such evaluations are costly and difficult to scale [43]. Proxy metrics offer a tractable, though imperfect, way to approximate long-term impact and anticipate how different mechanisms might perform. Ideally, a strong alternative review mechanism should correlate with—but not fully replicate—the outcomes of the current system, offering complementary perspectives and surfacing different strengths. As we show, our proposed pairwise comparison mechanism achieves this balance, aligning with current human judgments while introducing useful differentiation. Crucially, our goal is not to mimic existing outcomes, but to design mechanisms that add value.

Moreover, we believe the strength of human-led peer review lies not only in final decisions, but also in the human interactions it enables—feedback, discussion, and iterative refinement—that help authors strengthen their work. Through these interactions, reviewers and authors also engage in a shared evaluative practice that reinforces norms, builds trust, and helps individuals identify with the scholarly community [63, 19]. Our intention is not to automate or replace this process, but to broaden how we understand and support scholarly evaluation.

## 2   Related Work

**Challenges in Peer Review**   Peer review, while essential to scientific progress, faces numerous challenges in today's academic landscape. Reviewers often struggle with overwhelming workloads, further compounded by the absence of meaningful incentives and the inequitable distribution of review requests [7, 31]. In addition, implicit biases can influence evaluation [46, 47, 50], with studies showing that factors like author affiliation or gender may affect acceptance rates [59]. As research output continues to accelerate globally [5], these systemic challenges demand innovative solutions to preserve the integrity and effectiveness of scientific evaluation. Earlier works [48, 57, 56] introduce pairwise comparisons by incorporating author-provided rankings to calibrate noisy reviewer scores. In contrast, we fully replace paper-level scores with large-scale pairwise comparisons performed by LLM agents, without relying on human ratings or author input.

**LLMs for Peer Review**   Recent advancements in Large Language Models (LLMs) have sparked considerable interest in their potential to transform academic peer review processes [21, 62, 24, 23]. While existing studies [9, 22, 38] have demonstrated that LLMs can provide valuable feedback for research papers, exhibiting substantial overlap with reviews written by human reviewers, they often struggle with assigning reasonable ratings and making sound decisions, which raises concerns about their reliability for evaluative tasks [26, 58]. For instance, Niu et al. [35] showed that ChatGPT tends to grant acceptances: out of 1558 full paper evaluations, ChatGPT suggested 1243 acceptances compared to only 315 rejections, resulting in an acceptance rate of approximately 79.8%. Similarly, Zhou et al. [60] highlighted that LLMs could generate reasonable aspect scores (e.g. recommendation, soundness, originality) from human reviews but failed when given only the research paper. Lu et al. [27] improved the base LLM's decision-making process through prompting techniques such as self-reflection [42], few-shot examples [54] and response ensembling [53]. CycleReviewer [55] and DeepReview [61] enhanced LLM-based paper review via fine-tuning on high-quality review dataset.

In contrast, Less effort is made to explore new paradigms that fundamentally rethink how LLMs can participate in the academic review process.

## 3 Method

### 3.1 Framework Overview

We now describe our peer review mechanism (Figure 1) that transforms the traditional paper-level assessment process into a comparative ranking system through pairwise judgments. This approach builds on a key insight that evaluators are generally more reliable when deciding which of two items is better than when assigning each an absolute score [8]. By integrating a large number of pairwise judgments into a global ranking with the Bradley-Terry model, we could potentially mitigate the calibration inconsistencies and personal rating biases that plague conventional peer review systems.

The mechanism operates in three key stages: First, we collect pairwise paper comparisons from multiple LLM agents, where each agent evaluates a randomly sampled pair of papers and produces a binary preference judgment. Second, we apply the Bradley-Terry model to quantify each paper's relative quality through BT coefficients. Finally, we determine the optimal coefficients through maximum likelihood estimation and derive a complete ranking of all papers.

### 3.2 From Pairwise Comparisons to Rankings

Suppose we have $N$ paper candidates indexed by $i \in [N]$ and $M$ LLM agents participating in the evaluation process. We aim to establish a quality-based ranking through pairwise comparisons. The framework consists of the following steps:

1. **Pairwise Comparison**: Let $\mathcal{S} = \{(i, j) : i \neq j \text{ and } i, j \in [N]\}$ be all possible pairs of papers to compare. For each of $M$ agents, we randomly assign one pair $(i, j) \in \mathcal{S}$ to evaluate. Each agent then analyzes both papers and produces a binary preference judgment $y_{ij} \in \{0, 1\}$, where $y_{ij} = 1$ indicates that the agent prefers paper $i$ over paper $j$, and $y_{ij} = 0$ indicates the opposite preference. The full prompt used for pairwise judgment is provided in Supp Figure 7. Denote the collection of all chosen pairs as $\mathcal{A}$, where $|\mathcal{A}| = M$. We collect all comparison results into an observation set $\mathcal{O} = \{(i, j, y_{ij}) : (i, j) \in \mathcal{A}\}$, which contains triplets of the compared papers and their corresponding preference judgments from all agents. Since the size of $|\mathcal{S}|$ grows quadratically with $n$, we can only afford to assess a small fraction of all possible pairs, and hence it is challenging to recover the ranking under this scenario. Later in Section 4.2, we empirically examined how ranking quality scales with the number of LLM agents. The results reveal a clear scaling law, and we are able to recover a high-quality ranking with less than 2% samples of all possible pairs. [1]

2. **Bradley-Terry Model**: We employ the Bradley-Terry model [6] to recover the paper ranking from pairwise comparison results. The model provides stable statistical estimation from sparse comparisons, which is particularly suitable for our scenario. Specifically, it assumes each paper $i$ has an underlying quality score $\beta_i \in \mathbb{R}$, and the outcome of comparing papers $i$ and $j$ is a Bernoulli random variable, where the probability of paper $i$ beating $j$ is determined by a logistic function of the score difference $\beta_i - \beta_j$, i.e.,

$$p_{ij} := \mathbb{P}(y_{ij} = 1 | (i, j)) = \frac{e^{\beta_i}}{e^{\beta_i} + e^{\beta_j}} = \frac{1}{1 + e^{-(\beta_i - \beta_j)}} \tag{1}$$

To estimate these scores $\beta$ based on the outcome of observed pairwise comparisons, we utilize the maximum likelihood estimator on the Bradley-Terry model, which maximizes the log-likelihood function below:

$$\mathcal{L}(\beta) = \sum_{(i,j,y) \in \mathcal{O}} [y_{ij} \log(p_{ij}) + (1 - y_{ij}) \log(1 - p_{ij})]. \tag{2}$$

3. **Ranking Inference**: Once the scores are estimated, they can be used to rank all candidates in a descending order, with a higher score indicating "stronger" candidates with higher quality.

---

[1]Theorem 4 in [34] shows that if $m > 12n \log n$ pairs are sampled uniformly at random, then with high probability the estimate $\hat{\theta}$ satisfies $\|\hat{\theta} - \theta^*\| = O\left(n\sqrt{\frac{\log n}{m}}\right)$.

Since all our experiments were conducted using GPT-4o mini [36] for its balance of performance and computational cost, we will refer to our proposed mechanism as the GPT ranking system in the following sections.

# 4 Experiments

We now empirically validate whether our proposed pairwise-based ranking framework better identifies high-impact papers compared to conventional rating-based approaches.

## 4.1 Dataset Desciption

We collected papers from major ML conferences publicly available on OpenReview, including *ICLR*, *NeurIPS*, *CoRL*, and *EMNLP*. Papers were categorized into different decision outcomes (e.g., accepted vs. rejected, main vs. findings track, oral vs. poster, etc.), reflecting the quality assessments by the peer review process. The paper PDFs and corresponding decisions were retrieved using the OpenReview API (https://docs.openreview.net/). We used ScienceBeam [12] to extract the title, abstract, figure and table captions, and the main text. For each conference, we applied our method to the submitted papers and maintained the original distribution of papers across decision categories. Experiment setup details are available in the Appendix B.

## 4.2 Agent Scaling Boosts the Performance of GPT Ranking System

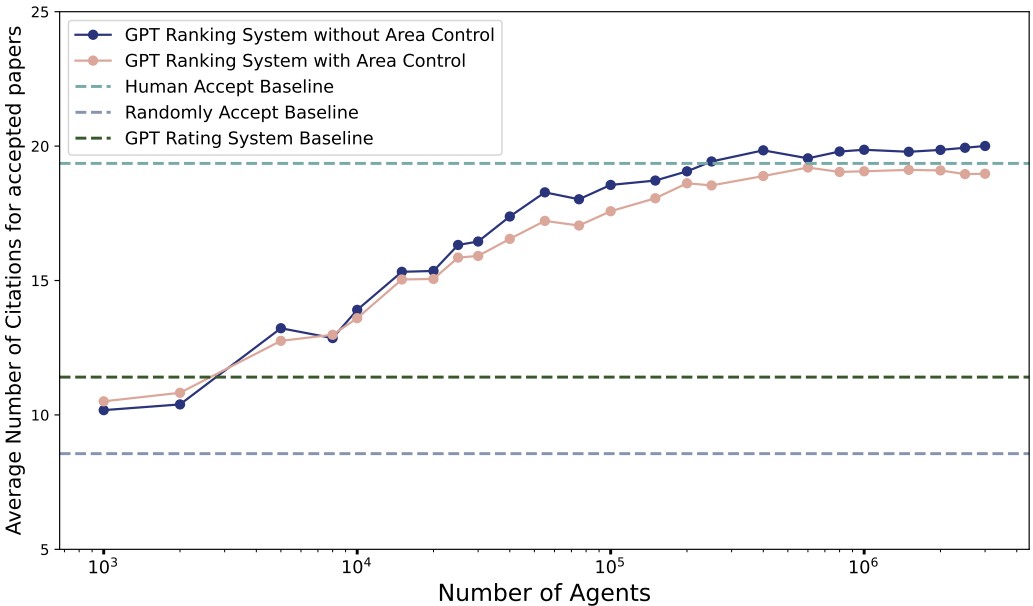

Figure 2: **Scaling of average citation counts for accepted papers by GPT ranking system with increasing pairwise comparisons**. We introduce two variants of our GPT ranking system: a basic version without area control, and one with area control, which ensures that the distribution of accepted papers across primary research areas matches that of human peer review. These two variants are compared against three baselines: (1) human acceptance decisions, (2) random acceptance based on the conference acceptance rate, and (3) GPT rating system (acceptance depends on the rating synthesized by a meta-reviewer from three independent GPT reviewers). As the number of comparisons increases, both variants steadily improve and approach human-level performance. At maximum scale, the GPT ranking system without area control achieves 20.00 average citations, while the version with area control reaches 18.97 average citations. These compare favorably to human acceptance decisions (19.36 average citations), and significantly outperform both random acceptance (8.56 average citations) and the GPT rating system baseline (11.41 average citations).

We investigate the scaling performance of our GPT ranking system with the number of agents, where each agent conducts a single pairwise comparison between randomly sampled papers from *ICLR 2024*. Figure 2 shows the overall academic impact (measured by their average citation counts) of the accepted papers as the number of agents increases. In particular, the system exhibits a steady increasing trend in average citations as the number of agents scales from around $10^3$ to $10^5$. At insufficient sample sizes (e.g., $10^3$ agents), the system performance approaches that of random acceptance. Since each paper receives too few comparisons, it fails to establish reliable relative rankings across the entire pool of 7158 papers. As the scale increases beyond $10^5$, the growth rate gradually slows down, with the average citation count eventually plateauing at approximately 20.

Given that the GPT ranking system demonstrates different acceptance patterns across research areas compared to humans (Section 4.5), we include two variants in our evaluation: one with area control, which maintains the same distribution of accepted papers across primary research areas as the human peer review system, and one without such controls. Nevertheless, they all yield comparable results to human-accepted papers at the convergence point. In the subsequent analysis, we focus on the GPT ranking system without area control as it better reflects the true properties of our system without deliberate controls.

We also compare our GPT ranking system with another baseline that follows a similar setup as prior works [27, 18]. This baseline, called the GPT rating system, involves three GPT reviewers independently generating reviews for each paper. Another agent then acts as a meta-reviewer to synthesize these individual reviews and provide a final rating (on a scale of 1 to 10). Acceptance decisions are made based on the rating provided by the meta-reviewer. However, it revealed a critical limitation: GPT-generated ratings exhibited less diversity, with most ratings concentrated around 6 and 7. Consequently, the system's ability to differentiate papers of varying quality was significantly constrained, leading to only minor improvements over the random baseline.

## 4.3  Discriminative Capability of GPT ranking system in Research Evaluation

With sufficient scale, our GPT ranking system approaches human-level performance in terms of average citation counts for accepted papers. However, beyond raw citation averages, a key question remains: does the GPT ranking system effectively distinguish between highly influential and less impactful papers in a manner comparable to human reviewers? To address this question, we analyzed the system's ability to discriminate papers across the entire citation distribution spectrum.

Across multiple top AI conferences (*ICLR 2024*, *EMNLP 2023*, *ICLR 2023*, and *CoRL 2023*) and under various decision conditions (accepted vs. rejected, main vs. findings track, oral vs. poster, etc.), papers ranked highly by our system consistently received more citations than those ranked lower (Figure 3, Supp Figures 8, 9). This pattern closely mirrors the citation advantages observed in human-selected papers across the same decision categories. The statistical significance of these differences (indicated by asterisks) further validates that our GPT ranking system can serve as reliable proxies for human peer review when identifying work likely to generate greater impact in the research community. The consistency of this pattern across different conferences and years suggests that as the system employs sufficient agents to conduct comprehensive pairwise comparisons, it could also capture the subtle quality signals that correlate with future scholarly influence.

## 4.4  Consistency Analysis between GPT Ranking System and Human Peer Review

We further examined the decision consistency between the GPT ranking system and conventional human peer review in *ICLR 2024* and *ICLR 2023*. As shown in Table 1, each paper was independently categorized by human reviewers and by the GPT ranking system into one of four decision categories — Oral, Spotlight, Poster, or Reject. Papers withdrawn after review releases were considered as rejected for this analysis.

Notably, 41.0% of the papers accepted by human reviewers in *ICLR 2024* were also accepted by the GPT ranking system (42.2% for papers accepted in *ICLR 2023*; Supp Table 4). The result aligns with findings from the *NeurIPS 2021* consistency experiment [3], where 48.0% of papers accepted by the first committee were also accepted by the second committee. This level of agreement between AI and human reviewers is roughly on par with the consistency observed between independent human review committees, suggesting that the GPT ranking system could offer valuable insights for identifying

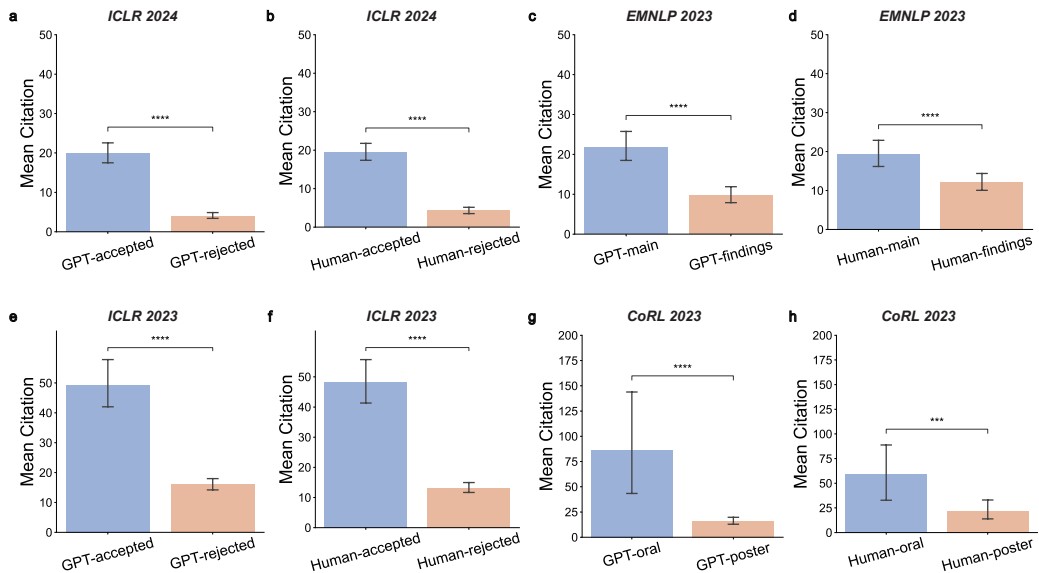

Figure 3: **Comparison of mean citation counts across multiple AI conferences under two-fold decisions (e.g., accepted vs. rejected, main vs. findings track, oral vs. poster).** The results consistently show that higher-tier papers (both GPT-selected and human-selected) receive substantially higher citation counts than lower-tier papers. This implies that our GPT ranking system can effectively distinguish more influential papers, performing comparably to human peer review system in identifying works likely to generate greater impact in the academic community. Error bars represent 95% confidence intervals. *$P < 0.05$, **$P < 0.01$, ***$P < 0.001$, and ****$P < 0.0001$.

Table 1: **Summary of recommendations for ICLR 2024 papers by two review systems: the human peer review (Human) and the GPT ranking system (GPT).** Each row represents humans' decision, while each column shows how the GPT ranking system categorized the same papers.

| Human\GPT | Oral | Spotlight | Poster | Reject |
|:---:|:---:|:---:|:---:|:---:|
| **Oral** | 6 | 11 | 28 | 41 |
| **Spotlight** | 10 | 32 | 130 | 191 |
| **Poster** | 29 | 116 | 556 | 1,085 |
| **Reject** | 41 | 204 | 1,072 | 3,606 |

potentially impactful papers that might otherwise be rejected due to the inherent randomness in the traditional peer review process.

### 4.5 Different Acceptance Patterns across Research Areas

We found notable disparities between the GPT ranking system and human peer review in their acceptance rates of several research areas (Figure 4, Supp Figure 10): GPT ranking system shows significantly higher acceptance rates for applied research areas such as robotics/autonomy applications (0.56 vs. 0.32) and societal considerations (0.51 vs. 0.30). In contrast, theoretical areas that received relatively high acceptance rates from human reviewers, such as learning theory (0.50 vs. 0.12) and optimization (0.31 vs. 0.12), had much lower acceptance rates from the GPT ranking system.

There are several ways to interpret these findings. First, LLMs are trained on massive web-crawled corpora rich in practical, application-oriented content. Consequently, these models may exhibit an inherent preference for studies that demonstrate immediate real-world applicability. Alternatively, large language models often struggle with complex mathematical reasoning [33], which could hinder their understanding of papers that require deep theoretical and mathematical rigor beyond memorized

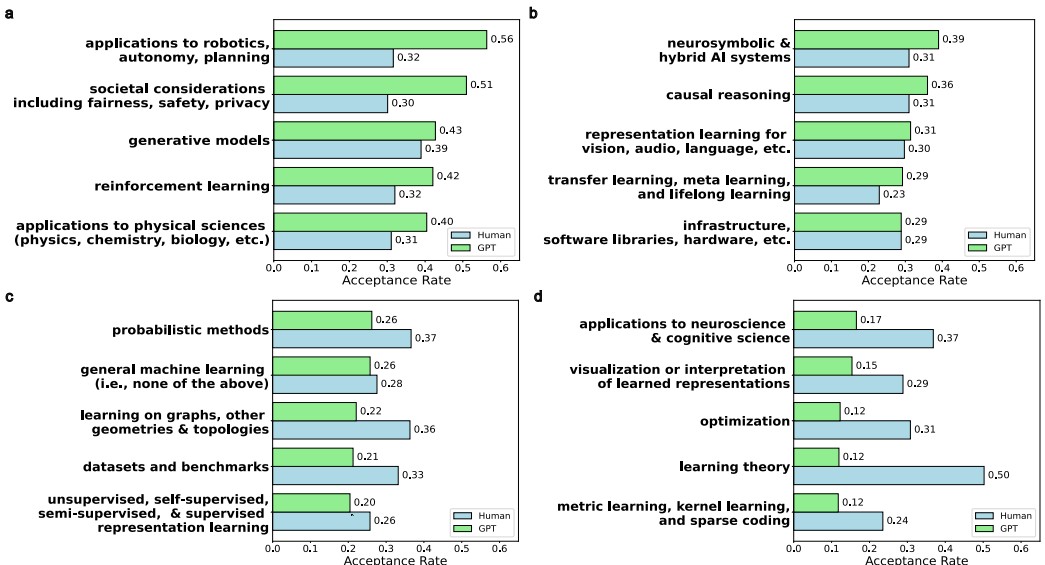

Figure 4: **Comparative acceptance rates of *ICLR' 24* papers by humans and GPT ranking system across research areas**. We sort areas by the GPT ranking system's acceptance rate from highest to lowest. The GPT ranking system exhibits noticeably higher acceptance rates in application-oriented fields compared to human reviewers, showing the most striking disparities in robotics (0.56 vs. 0.32) and societal considerations (0.51 vs. 0.30). In contrast, for more theoretical or methodologically focused areas, it assigns significantly lower acceptance rates than human reviewers. Learning theory demonstrates the largest gap, with acceptance rate at 0.12 versus humans at 0.50. The categorization is based on the 20 primary areas by *ICLR' 24*.

patterns. This limitation can result in a preference for research with more concrete implementations than abstract theoretical work. Future interdisciplinary research could explore these hypotheses.

### 4.6 Potential Bias against Novel Research Topics

We explored how our GPT ranking system influences the novelty of topics in selected papers. To measure the novelty of the topics, we embed each paper's abstract with OpenAI's text-embedding-3-small model, generating a vector representation for each abstract. We then compute the distance between each paper's vector and the closest neighbor within the abstracts from the same conference. A smaller distance indicates greater similarity between abstracts, thus lower novelty of the topic.

Through examination of papers categorized into top decision tiers by human reviewers and our GPT ranking system, our analysis revealed a consistent pattern (Figure 5): papers receiving higher rankings from the GPT system exhibited substantially smaller average distances to their nearest neighbors compared to papers selected by human reviewers. This might indicate that GPT exhibit a bias toward papers that cover topics similar to those in the existing literature, potentially undervaluing work that delves into novel research areas.

### 4.7 Impact on Academic Inequality

We also observed that our GPT ranking system tend to favor papers from established research institutions, potentially amplifying existing imbalances in the academic ecosystem. To quantify this imbalance, we track the first authors' affiliation distribution across papers presented in higher decision tiers. Using the Gini coefficient, a statistical measure commonly applied to income inequality, we quantified the degree of publication concentration across institutional affiliations.

We found a concerning pattern of institutional bias: the GPT ranking system consistently exhibited higher institutional inequality compared to humans, with significantly higher Gini coefficients across all conferences studied. The most significant disparity was observed in *ICLR 2023* and *ICLR 2024*.

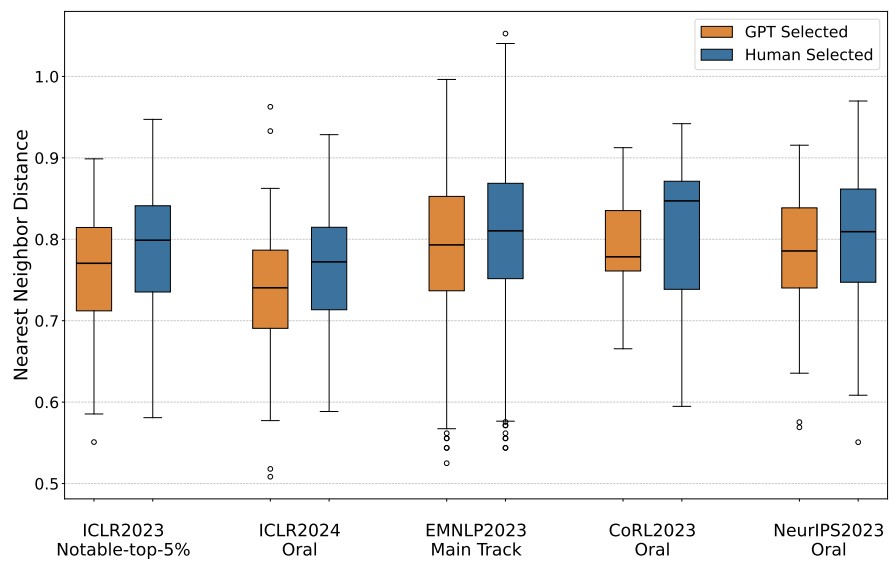

Figure 5: **Nearest neighbor distances of papers selected for top-tier acceptance by GPT ranking system versus human reviewers**. Higher distances indicate greater novelty. Human-selected papers (blue) consistently show higher nearest neighbor distances than GPT-selected papers (orange) across all conferences. The differences are statistically significant ($p < 0.05$) for ICLR 2023 Notable-top-5%, ICLR 2024 Oral, and EMNLP 2023 Main Track. For CoRL 2023 Oral and NeurIPS 2023 Oral, we did not observe statistically significant differences, which may be due to smaller sample sizes.

In *ICLR 2023*, 43.8% of GPT ranking system's top-tier papers came from 10 institutions compared to only 27.0% in human decisions, while in *ICLR 2024*, the gap persisted with 37.2% of GPT ranking system's top selections coming from 10 institutions versus 26.7% in human evaluations.

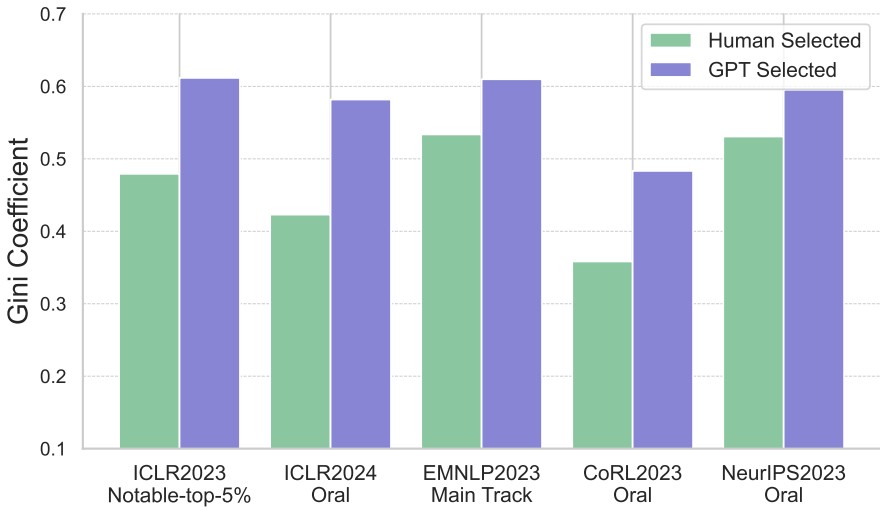

Figure 6: **Comparison of Gini coefficients for papers selected by humans versus GPT ranking system across conferences**. Larger Gini coefficients represent greater inequality or imbalance in the distribution. The consistently higher Gini coefficients for GPT-selected papers (purple bars) compared to human-selected papers (green bars) indicate that GPT ranking system exhibit greater institutional concentration, potentially favoring papers from established research institutions.

# 5 Conclusion

Existing LLM-assisted peer review efforts largely replicate traditional workflows, framing evaluation as absolute scoring followed by aggregation [18, 27, 49, 55, 61]. However, we argue that the scalability of LLMs open new opportunities to redesign scholarly evaluation structures rather than merely automate human processes. In this work, we introduce and explore a novel mechanism using LLM agents to evaluate academic papers through pairwise comparisons. Rather than assigning isolated scores, the system constructs a global ranking by aggregating local relative judgments between submissions. Through empirical experiments, we find that with sufficient scale, our system effectively identifies high-impact papers across multiple conferences, significantly exceeding traditional rating-based methods. The overlap between papers accepted by our system and those accepted by human reviewers aligns with human-human agreement levels observed in previous studies, suggesting potential value as a complementary tool in the review process.

At the same time, our analysis reveals important challenges. The system exhibits area-specific preferences that diverge from human reviewers, shows a measurable decline in topic novelty among selected papers, and concentrates acceptances among a smaller set of prestigious institutions than human-selected papers. These patterns suggest that while scalable LLM-driven evaluation systems hold promise, careful design will be critical to ensure they promote diversity, equity, and innovation rather than reinforcing existing hierarchies.

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

# A Discussion

## A.1 Limitations

In this study, we rely on citation counts as a proxy for academic impact. Although citation metrics are widely used and provide a quantifiable measure of influence, they are affected by numerous factors beyond quality, such as the visibility of research communities, prevailing research trends, and established reputations [25, 4, 51, 11]. More critically, papers selected by human reviewers inherently receive more visibility through conference presentations and proceedings. The fact that our GPT ranking system identified papers with comparable citation impact despite this disadvantage demonstrates a certain level of effectiveness in our approach. Nevertheless, this circularity problem makes it difficult to establish a truly independent measure of quality. To further verify the robustness of our proposed method, we conducted experiments using alternative measures. Specifically, we measured the Spearman correlation between our system's output and human review scores. In both ICLR 2023 and ICLR 2024, the BT scores produced by our mechanism show moderate correlation with the average human ratings—27% and 24%, respectively. This indicates that our system aligns with signals in the current review process, while not fully replicating its outcomes. We believe this complementary nature is essential for identifying potential improvements to existing peer review systems.

Furthermore, while we exclusively used GPT-4o mini for our large-scale experiments due to its balance of performance and computational cost, we acknowledge that there are other diverse LLMs capable of academic assessment, and they might yield substantially different results. A more comprehensive evaluation across multiple models would provide better insight into the generalizability of our method and findings. In Appendix C.4, we tested two additional models beyond GPT-4o mini: Gemini 2.0 Flash and Claude-3-Haiku-20240307. We scaled the number of comparisons to over $10^6$ and found that, in both cases, our pairwise comparison framework performs significantly better than the GPT rating system baseline. Specifically, papers selected by Gemini 2.0 Flash receive an average of 18.3 citations (vs. 11.4), while those chosen by Claude-3-Haiku average 16.8 (vs. 11.4). These results suggest that the advantage of our approach is robust across different LLMs.

Finally, while our current work constructs a single global ranking, future extensions could explore personalized or multi-objective evaluation systems that explicitly account for epistemic diversity and evolving community goals.

## A.2 Expanding Peer Review through Pairwise Evaluation

Beyond the immediate results, our framework and explorations open broader directions for redesigning scholarly review systems.

First, because pairwise comparisons produce local relative judgments that can be incrementally aggregated, our approach naturally lends itself to continuous evaluation. Rather than operating within a fixed submission and decision timeline, conferences could maintain an ongoing review pipeline, where new papers are progressively compared against the existing submission pool. Such a dynamic process could allow for rolling acceptances, faster feedback loops, and better accommodation of late-breaking research.

Second, pairwise evaluation offers distinct advantages for emerging or interdisciplinary fields where traditional scoring rubrics are poorly defined or difficult to establish. Absolute scoring requires consensus on evaluation criteria and careful calibration across reviewers, which can be challenging in fast-moving or nascent research areas. Comparative judgments, by focusing on relative rather than absolute assessments, can surface high-potential work even when shared evaluation norms are still evolving, making the system more adaptable to domains where innovation resists rigid checklist-based quantification.

Third, the flexibility of the aggregation process suggests opportunities for personalized and diversified peer review. Different weighting schemes could prioritize novelty, interdisciplinarity, methodological rigor, or other dimensions based on the goals of specific conferences, tracks, or even reviewer communities. In principle, such mechanisms could allow peer review to better reflect the heterogeneous values of different research communities, moving beyond the one-size-fits-all model [15] currently dominant in scientific publishing.

Finally, an important direction for future exploration lies in integrating human expertise and LLM evaluations [41, 29]. Rather than viewing LLM-based and human-based assessments as competing alternatives, hybrid models could combine the strengths of both: leveraging LLMs for large-scale, consistent pairwise comparisons while relying on human reviewers to provide deeper qualitative insights, assess boundary cases, and adjudicate particularly novel or interdisciplinary submissions. Designing effective protocols for human-AI collaboration in peer review could further enhance both the scalability and the fairness of the evaluation process.

Together, these directions highlight how reframing peer review around relative comparisons, enriched by human-AI collaboration, could support a more scalable, inclusive, and adaptive scholarly communication ecosystem.

## B    Experimental Details

### B.1    Dataset Details

Here we include additional details on the datasets used for our experiments.

Table 2: **Academic paper data from major ML conferences.**

| Conference | Decision Types | # of Papers |
|---|---|---|
| **ICLR 2023** | • Notable-top-5% | 89 |
| | • Notable-top-25% | 281 |
| | • Poster | 1,193 |
| | • Reject (Withdrawn submissions included) | 3,303 |
| **ICLR 2024** | • Accept (Oral) | 86 |
| | • Accept (Spotlight) | 363 |
| | • Accept (Poster) | 1,786 |
| | • Reject (Withdrawn submissions included) | 4,923 |
| **NeurIPS 2023** | • Accept (Oral) | 67 |
| | • Accept (Spotlight) | 374 |
| | • Accept (Poster) | 2,748 |
| **EMNLP 2023** | • Accept-Main | 975 |
| | • Accept-Findings | 993 |
| **CoRL 2023** | • Accept (Oral) | 32 |
| | • Accept (Poster) | 166 |

### B.2    Implementation Details

We use the official OpenAI's Batch API for GPT-4o mini and set temperature as 0 during pairwise comparison. The number of API calls and estimated cost for each simulated conference is as follows:

Table 3: **API usage and estimated costs for pairwise comparison across each conference.** *EMNLP 2023* and *CoRL 2023* used all available pairs, while others were randomly downsampled to 3M pairs.

| Conference | # of API Calls | Estimated Cost (USD) |
|---|---|---|
| **ICLR 2023** | 3,000,000 | $\sim 1,350$ |
| **ICLR 2024** | 3,000,000 | $\sim 1,350$ |
| **NeurIPS 2023** | 3,000,000 | $\sim 1,350$ |
| **EMNLP 2023** | 3,871,056 | $\sim 1,700$ |
| **CoRL 2023** | 39,006 | $\sim 17$ |

## B.3 LLM prompts used in the study

```
Please act as an impartial judge and evaluate the quality of the
    following two papers. As the area chair for a top ML conference,
    you can only select one paper. Start with a brief meta-review/
    reasoning of the pros and cons for each paper (two sentences), and
     then provide your choice in a binary format. Start with a brief
    meta-review/reasoning of the pros and cons for each paper,
    focusing on novelty, significance, clarity, methodology, and
    practical implications. Be very critical and do not be biased by
    what the author claimed. Finally, provide your choice in a binary
    format.

Please provide your analysis in JSON format.

### Paper 1:
Submission Title: {title}

'''
Abstract: {abstract}

Figures Captions:
{figure_and_table_captions}

Main:
{main_content}
'''

### Paper 2:
Submission Title: {title}

'''
Abstract: {abstract}

Figures Captions:
{figure_and_table_captions}

Main:
{main_content}
'''

Your JSON output should look like this:

{{
    "paper_1_review": "Your meta-review and reasoning for paper 1",
    "paper_2_review": "Your meta-review and reasoning for paper 2",
    "chosen_paper": "paper_1 or paper_2"
}}
"""
```

Figure 7: Example prompt for pairwise comparison.

# C Additional Results

## C.1 Discriminative Capability of GPT ranking system in Research Evaluation

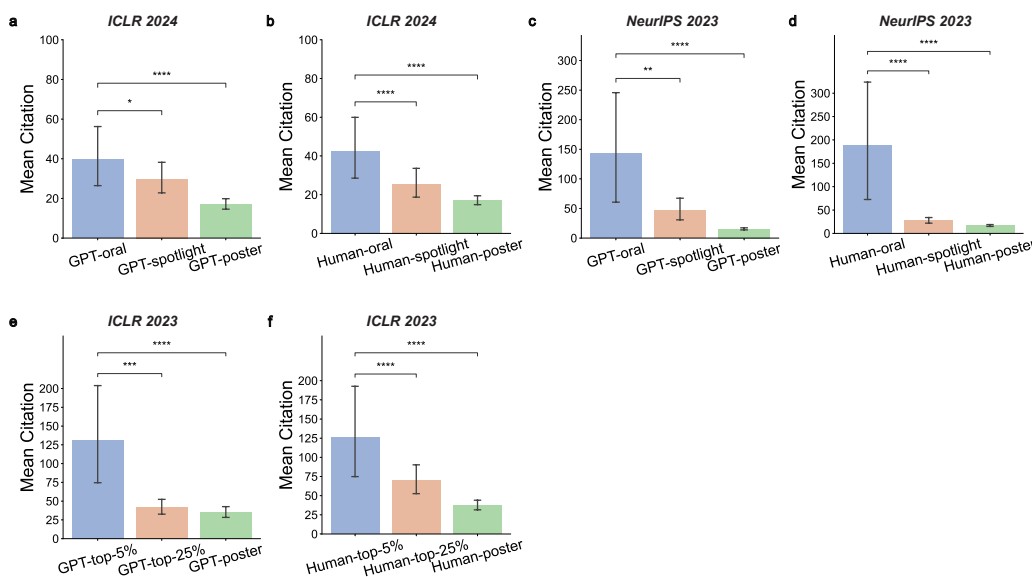

Figure 8: **Comparison of mean citation counts across multiple AI conferences under fine-grained decisions in accepted papers.** The results consistently show that higher-tier papers (both GPT-selected and human-selected) receive overall higher influence than lower-tier papers. Error bars represent 95% confidence intervals. *P < 0.05, **P < 0.01, ***P < 0.001, and ****P < 0.0001.

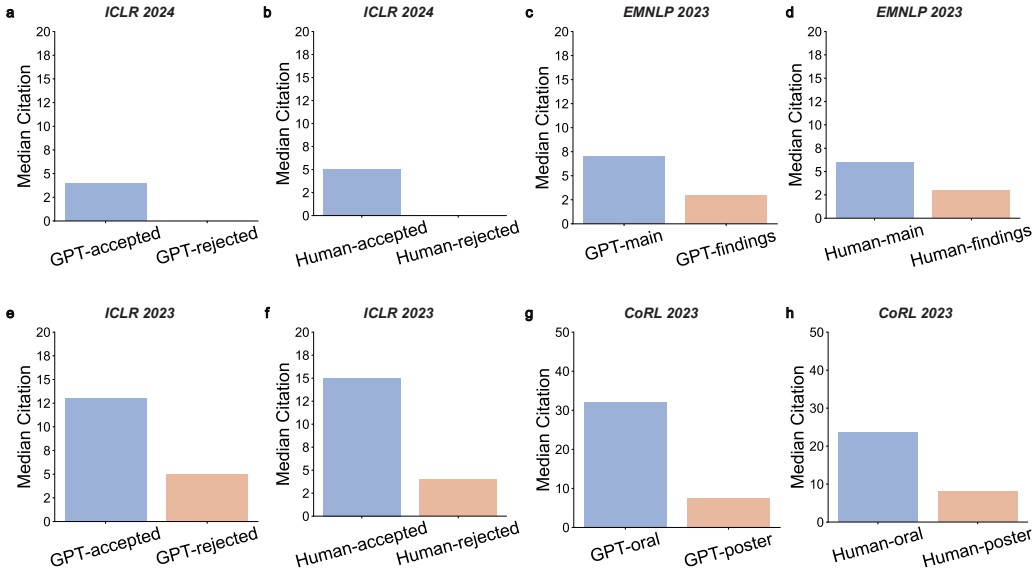

Figure 9: **Comparison of median citation counts across multiple AI conferences under two-fold decisions (e.g., accepted vs. rejected, main vs. findings track, oral vs. poster).** The results consistently show that higher-tier papers (both GPT-selected and human-selected) receive overall higher influence than lower-tier papers.

## C.2 Consistency Analysis between GPT Ranking System and Human Peer Review

Table 4: **Summary of recommendations for ICLR 2023 papers by two review systems: the human peer review (Human) and the GPT ranking system (GPT).** Each row represents humans' decision, while each column shows how the GPT ranking system categorized the same papers.

| Human\GPT | Notable top 5% | Notable top 25% | Poster | Reject |
|---|---|---|---|---|
| **Notable top 5%** | 10 | 9 | 26 | 44 |
| **Notable top 25%** | 11 | 27 | 95 | 148 |
| **Poster** | 29 | 93 | 360 | 711 |
| **Reject** | 39 | 152 | 712 | 2,400 |

## C.3 Different Acceptance Patterns across Research Areas

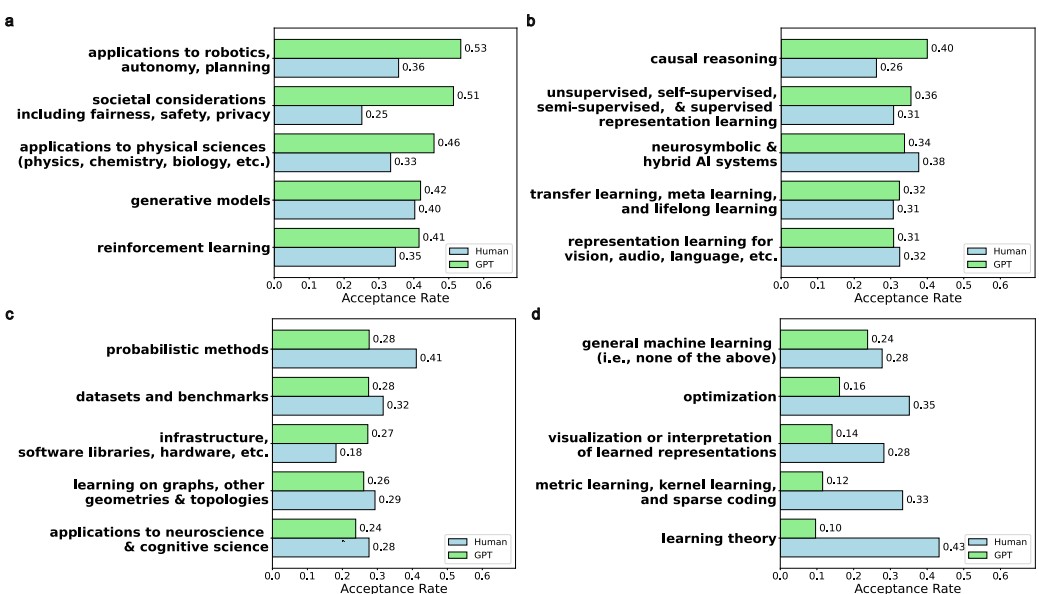

Figure 10: **Comparative acceptance rates of *ICLR' 23* papers by humans and GPT ranking Systems across research areas**. We sort areas by the GPT ranking system's acceptance rate from highest to lowest. The GPT ranking system exhibits noticeably higher acceptance rates in application-oriented fields compared to human reviewers, showing the most striking disparities in robotics (0.53 vs. 0.36) and societal considerations (0.51 vs. 0.25). In contrast, for more theoretical or methodologically focused areas, it assigns significantly lower acceptance rates than human reviewers. Learning theory demonstrates the largest gap, with acceptance rate at 0.10 versus humans at 0.43.

## C.4 Using other LLMs as agents

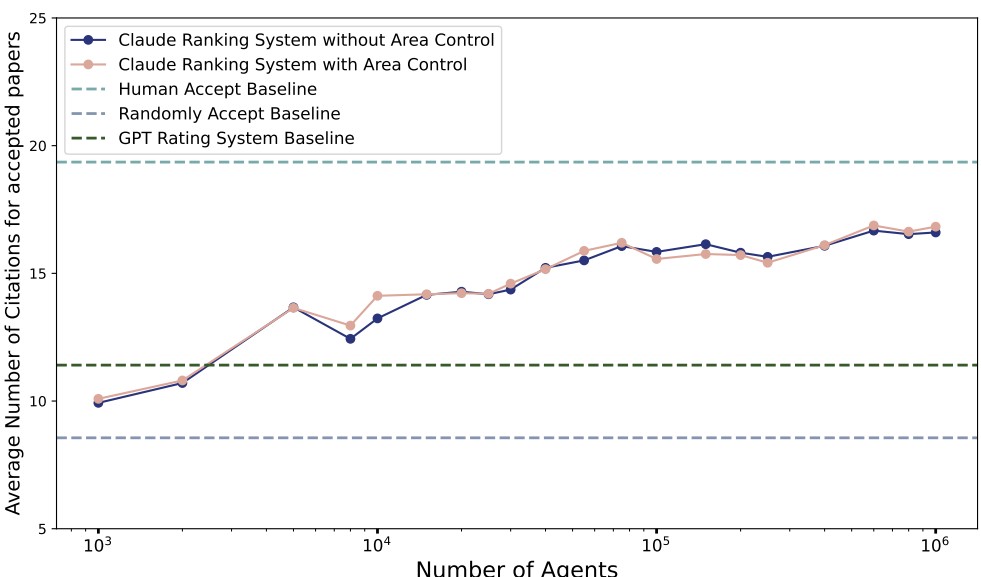

Figure 11: **Scaling of average citation counts for accepted papers by Claude ranking system with increasing pairwise comparisons**. We observe a similar temporal scaling pattern as in the GPT ranking system, with citation counts increasing as the number of pairwise comparisons grows.

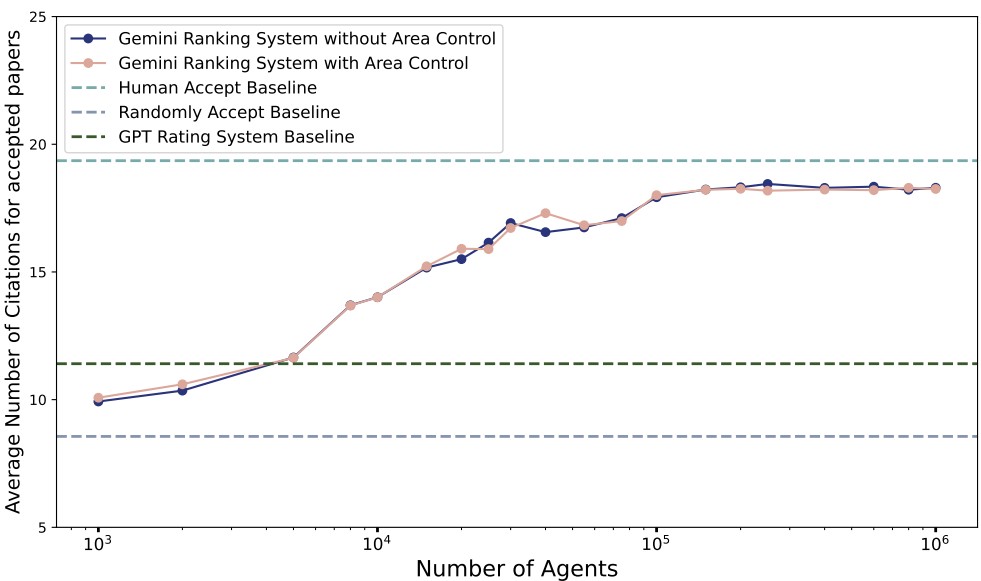

Figure 12: **Scaling of average citation counts for accepted papers by Gemini ranking system with increasing pairwise comparisons**. We observe a similar temporal scaling pattern as in the GPT ranking system, with citation counts increasing as the number of pairwise comparisons grows.

