# OpenReview forum: "From Replication to Redesign: Exploring Pairwise Comparisons for LLM-Based Peer Review"
_NeurIPS.cc/2025/Conference — NeurIPS 2025 poster_

### Official Review · Reviewer_gH5a · 2025-06-19

**Clarity:** 3
**Significance:** 3
**Originality:** 3
**Rating:** 5
**Confidence:** 4

**Summary:**

In this paper, the authors propose a novel peer review mechanism that replaces traditional per-paper scoring with pairwise comparisons conducted by LLM agents. Instead of assigning absolute ratings, each agent compares two papers and outputs a binary preference. These comparisons are aggregated using the Bradley-Terry model to generate a global ranking of submissions. The approach is empirically validated on papers from major ML and NLP conferences, showing improved alignment with citation-based impact metrics compared to standard rating-based LLM review pipelines. Notably, the authors identify tradeoffs in the system: while it scales well and improves consistency, it tends to favor applied research over theoretical work, reduces topic novelty, and amplifies institutional inequality—highlighting both the promise and limitations of LLM-based evaluation systems.

**Questions:**

See weaknesses.

**Ethical Concerns:**

["NO or VERY MINOR ethics concerns only"]

**Final Justification:**

I have read the authors' rebuttal. My initial review was highly positive, and I am maintaining my recommendation to accept. The paper's core strength lies not only in its novel and well-validated pairwise comparison method but also in its crucial and transparent analysis of the system's inherent biases. As I did not raise any major concerns, my positive assessment stands unchanged.

**Limitations:**

See weaknesses.

**Paper Formatting Concerns:**

N/A.

**Quality:**

3

**Strengths And Weaknesses:**

The paper is well written, with clear pointers to mathematical details where appropriate.

The paper proposes a fundamental shift from traditional rating-based evaluation to a pairwise ranking system using LLM agents.

The authors conduct comprehensive experiments across multiple top-tier conference datasets, showing that the proposed method achieves acceptance patterns and impact levels comparable to those of human reviewers.

Beyond reporting performance metrics, the paper analyzes biases in topic novelty, institutional concentration, and research area preferences, contributing to the ongoing discussion of fairness and diversity in AI-assisted evaluation systems.

---

> ### Author Rebuttal · Authors · 2025-07-30
>
> Thank you for your feedback and the time you invested in reviewing our manuscript. We are delighted that you found our paper to be well-written, clear, and compelling. If there are specific concerns or areas that were unclear, we would be happy to address them in detail.

---

### Official Review · Reviewer_tDGv · 2025-06-30

**Clarity:** 4
**Significance:** 2
**Originality:** 2
**Rating:** 3
**Confidence:** 4

**Summary:**

## Update after rebuttal

Given that the authors’ responses have addressed some of my concerns, I raise the Quality score from 2 to 3; the Significance score from 1 to 2; and the overall Rating from 2 to 3.

I still hold a negative view on the novelty, but it is subjective -- defer to the Area Chair for the final judgment.

## Summary

This paper explores the use of LLMs to automate the peer review process. Instead of following the traditional approach of having an LLM evaluate each paper individually, the authors use LLMs to perform **pairwise comparisons between papers**. The comparison results are then aggregated using the Bradley-Terry model to recover a global ranking of the papers. This approach is similar to [1] and its subsequent works, with the key difference being that LLMs, rather than authors, perform the pairwise comparisons.

In the experimental section, the authors use the number of citations a of paper as the proxy for its quality and evaluate the performance of the proposed system (referred to as the "GPT ranking system") against human reviewers. The results show that the average citation count of papers accepted by the GPT ranking system is comparable to that of papers accepted by human reviewers.

In addition to the main experimental results, the authors also conducted further analyses, such as the discriminative capability of the GPT ranking system and the consistency analysis between the GPT ranking system and human peer review.

[1] Su W. *You are the best reviewer of your own papers: An owner-assisted scoring mechanism*. Advances in Neural Information Processing Systems, 2021, 34: 27929–27939.

**Questions:**

See Questions in the "Weaknesses" paragraph.

**Ethical Concerns:**

["NO or VERY MINOR ethics concerns only"]

**Final Justification:**

## Update after rebuttal

Given that the authors’ responses have addressed some of my concerns, I raise the Quality score from 2 to 3; the Significance score from 1 to 2; and the overall Rating from 2 to 3.

I still hold a negative view on the novelty, but it is subjective -- defer to the Area Chair for the final judgment.

**Limitations:**

yes

**Quality:**

3

**Strengths And Weaknesses:**

## Overall Merit

I like this work, especially its experiments. However, the idea presented is not quite novel, and the performance of the proposed method does not significantly exceed the baseline -- human reviewers. I would classify this paper as a solid but ordinary piece of work, and I believe it does not meet the bar for acceptance at NeurIPS.


## Strengths

- The experimental section is very thorough and covers most of the key questions that readers may have.
- The organization and writing of the paper are excellent and very satisfying.


## Weaknesses

- The idea of using pairwise comparison to improve the peer review process is not entirely novel—see [1] and its follow-up works.
- Whether LLMs are truly capable of performing academic paper pairwise comparisons is not very intuitive. Although the authors provide the prompt in the appendix, I would prefer to see a more detailed explanation and analysis in the main text.
- There are many methods available for recovering rankings from pairwise comparisons. The authors might consider going beyond the Bradley-Terry model.
- I would like the authors to add a paragraph analyzing how much noise LLM agents introduce when making pairwise comparisons. Since multiple LLM agents were used for comparing each pair of papers (I'm not sure the source of randomness), this analysis likely does not require additional experiments. I would be very interested to see a comparison of the noise from LLM agents vs. human reviewers on the same set of papers.
- The paper only uses GPT as the LLM agent. I would like to see experimental results using other LLMs as agents, not just GPT.
- I disagree with using absolute citation count as the evaluation metric for paper quality. Citation counts are heavily influenced by research domain, so I believe that the comparison between "GPT Ranking System without Area Control" and human reviewers does not provide meaningful insight.
- The proposed method, "GPT Ranking System," achieves performance comparable to human reviewers in the experiments. However, this result is obtained by using a large number of LLM agents, whereas human reviewers typically rely on just 3 to 5 reviewers per paper. This suggests that the actual performance of the proposed method may not be as strong as it appears.

---

> ### Author Rebuttal · Authors · 2025-07-30
>
> Thank you very much for your thoughtful feedback! We have added several detailed explanations and new experiments to address each of your comments. Please let us know if you have any further questions.
>
> > The idea of using pairwise comparison to improve the peer review process is not entirely novel.
>
> Thank you for the comment. While we acknowledge the contributions of prior work on using pairwise comparison in peer review, our method differs in several key ways:
>
> **Shift from Human Ratings to LLM-Driven Comparative Judgments**: Earlier works [1-3] introduce pairwise comparisons by incorporating author-provided rankings to calibrate noisy reviewer scores. In contrast, we fully replace paper-level scores with large-scale pairwise comparisons performed by LLM agents, without relying on human ratings or author input.
>
> **Rethinking the Review Process Design**: While previous studies often aim to improve score accuracy within the traditional workflow (both human and LLM-based peer review), our approach reimagines the decision-making layer itself. We propose a fully comparative mechanism where ranking is derived entirely from relative assessments via the Bradley-Terry model.
>
> We believe this transition, from replication to redesign, is a meaningful and novel contribution to the literature on LLM-based peer review. We have modified the manuscript to highlight these differences.
>
> > Whether LLMs are truly capable of performing academic paper pairwise comparisons is not very intuitive. Although the authors provide the prompt in the appendix, I would prefer to see a more detailed explanation and analysis in the main text.
>
> Thank you for the suggestion. Our approach builds on recent findings that strong LLM judges can reliably perform pairwise comparisons that closely align with human preferences [4]. In addition, prior studies have shown that evaluators are generally more reliable when deciding which of two items is better than when assigning each an absolute score [5]. We also present extensive empirical results in the Experiments Section demonstrating that the system consistently identifies higher-impact papers across multiple conferences and decision settings.
>
> In the revised manuscript, we have expanded the explanation of our pairwise evaluation setup and discussed the need for future human studies to assess whether LLMs can reliably compare academic papers.
>
> > There are many methods available for recovering rankings from pairwise comparisons. The authors might consider going beyond the Bradley-Terry model.
>
> Thank you for the suggestion. While we agree that there exist many approaches for recovering rankings from pairwise comparisons, we chose the Bradley–Terry model because it offers a simple, interpretable, and statistically grounded method, and has been widely adopted in the machine learning community. For example, Chatbot Arena uses the BT model to aggregate pairwise judgments from human voters, and Direct Preference Optimization (DPO) builds on a similar formulation to learn from binary preferences. We believe it is a strong and practical choice, and leave exploration of alternative aggregation methods to future work.
>
> > I would like the authors to add a paragraph analyzing how much noise LLM agents introduce when making pairwise comparisons. Since multiple LLM agents were used for comparing each pair of papers (I'm not sure the source of randomness), this analysis likely does not require additional experiments. I would be very interested to see a comparison of the noise from LLM agents vs. human reviewers on the same set of papers.
>
> Thank you for the opportunity to clarify. We clarified that in our method, we uniformly random-sample from all pairs and then use the Bradley–Terry model to recover global rankings from sparse but representative observations. We empirically show that the required sample size for a high-quality ranking is significantly smaller than the total number of possible combinations (Figure 2). The scaling performance of our framework also allows us to study how noise in individual pairwise judgments affects the overall ranking quality. As a result, each paper pair is evaluated only once with high probability. This sampling process helps minimize the potential for bias or noise that could arise from repeated evaluations of the same paper pairs.
>
> While we have not directly compared the noise introduced by LLM agents and human reviewers on the same set of papers, we agree that such a comparison would provide valuable insights into the relative consistency of both methods. We suggest that future work explore this comparison to better understand how noise in LLM-based pairwise comparisons compares to human evaluation.
>
> > The paper only uses GPT as the LLM agent. I would like to see experimental results using other LLMs as agents, not just GPT.
>
> Thank you for the suggestion. We have conducted experiments to test two additional models other than GPT-4o mini: Gemini 2.0 Flash and Claude-3-haiku-20240307. We scaled the number of comparisons to over $10^6$ and found that, in both cases, our pairwise comparison framework performs significantly better than the GPT rating system baseline. Specifically, papers selected by Gemini 2.0  Flash receive an average of 18.3 citations (vs 11.4), while those chosen by Claude‑3‑Haiku average 16.8 (vs 11.4). These results suggest that the advantage of our approach is robust across different LLMs.
>
> > I disagree with using absolute citation count as the evaluation metric for paper quality. Citation counts are heavily influenced by research domain, so I believe that the comparison between "GPT Ranking System without Area Control" and human reviewers does not provide meaningful insight.
>
> Thank you for raising this important point. We acknowledge that citation counts, while widely used, are an imperfect reflection of research quality. As described in Appendix A.1, they are affected by numerous factors beyond quality, such as the visibility of research communities, research domain, etc.
>
> To further verify the robustness of our proposed method, we conducted experiments using alternative measures. Specifically, we measured the Spearman correlation between our system’s output and human review scores. In both ICLR 2023 and ICLR 2024, the BT scores produced by our mechanism show moderate correlation with the average human ratings—27% and 24%, respectively. This indicates that our system aligns with signals in the current review process, while not fully replicating its outcomes. We believe this complementary nature is essential for identifying potential improvements to existing peer review systems.
>
> Moreover, we  added explicit clarification to emphasize that these metrics are used as a practical means of evaluation: “ It is important to note that while we use proxy metrics—such as future citation counts—to evaluate our proposed mechanism, this represents an exploratory and pragmatic step rather than an ideal. The gold standard would be direct human evaluation of how well a mechanism supports the selection and improvement of high-quality research, but such evaluations are costly and difficult to scale . Proxy metrics offer a tractable, though imperfect, way to approximate long-term impact and anticipate how different mechanisms might perform. Ideally, a strong alternative review mechanism should correlate with—but not fully replicate—the outcomes of the current system, offering complementary perspectives and surfacing different strengths. As we show, our proposed pairwise comparison mechanism achieves this balance, aligning with current human judgments while introducing useful differentiation.”
>
> > The proposed method, "GPT Ranking System," achieves performance comparable to human reviewers in the experiments. However, this result is obtained by using a large number of LLM agents, whereas human reviewers typically rely on just 3 to 5 reviewers per paper. This suggests that the actual performance of the proposed method may not be as strong as it appears.
>
> Thank you for the comment. We agree that our system utilizes a larger number of LLM agents than the typical number of human reviewers per paper. However, we believe this reflects a fundamental trade-off between time, cost, and scalability. Conventional peer review is an expensive and time-consuming process—each human review takes several hours on average and incurs substantial labor and opportunity costs [6-7]. This challenge is becoming increasingly pronounced due to the rapid growth of research output and a persistent shortage of qualified reviewers.
>
> In contrast, our GPT ranking system leverages the scalability of LLM agents to perform thousands of pairwise comparisons, enabling the construction of a global ranking that would be impractical for human committees. This demonstrates the potential for scalable, low-cost review mechanisms, which may complement traditional peer review in settings where efficiency and volume are prioritized. Rather than replicating the conventional peer review process, our goal is to explore how LLM-based systems can be designed to generate effective, reliable assessments.
>
> [1] Su, Weijie. "You are the best reviewer of your own papers: An owner-assisted scoring mechanism." NeurIPS 2021
>
> [2] Yan et al. "Isotonic mechanism for exponential family estimation in machine learning peer review." J. R. Stat. Soc. Ser. B Methodol 2025
>
> [3] Wu et al. "A truth serum for eliciting self-evaluations in scientific reviews." arXiv:2306.11154 (2023)
>
> [4] Zheng et al. "Judging llm-as-a-judge with mt-bench and chatbot arena." NeurIPS 2023
>
> [5] Carterette et al. "Here or there: Preference judgments for relevance." ECIR 2008, Springer
>
> [6] LeBlanc et al. "Scientific sinkhole: estimating the cost of peer review based on survey data with snowball sampling." Res. Integr. Peer Rev. 2023
>
> [7] Aczel et al. "A billion-dollar donation: estimating the cost of researchers’ time spent on peer review." Res. Integr. Peer Rev. 2021

---

> > ### Comment · Reviewer_tDGv · 2025-08-02
> > **Reviewer Response**
> >
> > ## Further Comments
> >
> > > The idea of using pairwise comparison to improve the peer review process is not entirely novel.
> >
> > Literally, using LLMs to conduct pairwise evaluations is simpler than relying on authors to perform pairwise evaluations of papers. The reasons include the fact that the latter requires accounting for strategic behavior and difficulties in recovering paper rankings. **Although I still hold a (subjectively) negative view regarding the novelty of this paper, I observed that the other reviewers generally recognize its novelty, so perhaps I should not insist too strongly on this point.**
> >
> > > About how much noise LLM agents introduce when making pairwise comparisons.
> >
> > Since the authors claim that "each paper pair is evaluated only once with high probability," they are unable to provide an analysis of the noise introduced by LLMs during pairwise comparisons. I don't think this is critical, but I am genuinely curious about this. If the authors could conduct a small-scale experiment—even just repeatedly evaluating a few hundred paper pairs to analyze the LLM noise—it would be highly valuable.
> >
> > > About experimental results using other LLMs as agents, not just GPT.
> >
> > The authors have provided the results I was looking for, demonstrating that their method is effective across a broad range of LLMs.
> >
> > > About absolute citation count as the evaluation metric for paper quality.
> >
> > I am satisfied with the additional results the authors provided. I hope a detailed discussion of this will be included in the manuscript.
> >
> > ## Revisions (if authors do not have further feedback)
> >
> > Given that the authors’ responses have addressed some of my concerns, I will raise the Quality score from 2 to 3; the Significance score from 1 to 2; and the overall Rating from 2 to 3.
> >
> > I still hold a negative view on the novelty, but it is subjective -- defer to the Area Chair for the final judgment.

---

> > > ### Author Response · Authors · 2025-08-02
> > >
> > > Thank you for your response! We appreciate your feedback and understand your decision on the rating. However, we believe there may be some misunderstandings that we'd like to address: our primary goal is to highlight the opportunities and challenges of peer review mechanism design in the era of LLMs. The main contribution of our work lies in demonstrating that LLMs, when used as comparative judges, can enable a scalable and effective alternative to **existing LLM-based pipelines** that closely mimic traditional human review processes. We hope this will encourage the community to explore new LLM-based review paradigms, rather than merely replicating the current imperfect human review systems.
> > >
> > > We sincerely appreciate your engagement and your helpful suggestions, especially your comment regarding the noise in LLM judgments. Following your suggestion, we conducted an additional small-scale experiment where we randomly sampled 1,000 paper pairs and evaluated each pair 50 times using GPT-4o mini with temperature set to 1. We then computed the **consistency rate** for each pair, defined as the proportion of evaluations that agreed with the majority preference. Averaged across all 1,000 pairs, we observed a mean consistency rate of 81.1%, suggesting that LLMs exhibit reasonably stable judgments overall despite potential noise from sources such as temperature randomness and prompt sensitivity.
> > >
> > >
> > > We hope this clarification addresses your concerns and better explains our work. Thank you again for your time and consideration.

---

> > > > ### Comment · Reviewer_tDGv · 2025-08-03
> > > > **Reviewer Response**
> > > >
> > > > I thank the author for supplementing the experiment. I have no further questions.

---

### Official Review · Reviewer_rL5P · 2025-07-02

**Clarity:** 3
**Significance:** 2
**Originality:** 3
**Rating:** 5
**Confidence:** 4

**Summary:**

This paper proposes a new LLM-based pairwise comparison mechanism to rank manuscripts. The LLMs are prompted to give a meta-review + reasoning for each paper, followed by a binary indicator to show the preferred manuscript. Based on these scores, the authors estimate a Bradley-Terry model to recover the manuscript ranking from these pairwise comparisons. In their experiments, the authors use OpenReview data, including ICLR, NeurIPS, CoRL, and EMNLP, and GPT-4o mini (context window: 128K tokens). They empirically find that the theoretical burden of the O(n^2) scaling does not apply as with less than 2% of the samples a high-quality ranking is recovered. The authors validate the LLM-based ranking via citation metrics and human reviewer decisions (e.g., accepted vs. rejected, main vs. findings track, oral vs. poster, etc.). Finally, the authors also show that this mechanism tends to reduce novelty (measured via distance in word embeddings) and increases institutional imbalance.

**Questions:**

- Are the differences in the novel research topics statistically significant? There seems to be a large overlap between the distributions.
- Do the authors have any suggestions on how to incorporate responses to the meta-reviews? The back and forth (e.g., clarifications, additional experiments) between reviewers and authors is an important part of the review process which is omitted using this approach.

**Ethical Concerns:**

["NO or VERY MINOR ethics concerns only"]

**Final Justification:**

Update after rebuttal:
The overall score has been increased to 5 and quality to 3. The authors have assessed statistical significance + provided some theoretical bounds on the number of necessary pairs to obtain a qualitative ranking.

**Limitations:**

The authors discuss the limitations in Appendix Section A1.

**Paper Formatting Concerns:**

No concerns.

**Quality:**

3

**Strengths And Weaknesses:**

Strengths:
- I like how the authors do not try to replicate the existing review process, but argue that LLMs may offer the opportunity to redesign the entire process, which they argue mostly reflects historical compromises rather than principled design choices.
- The authors propose a clear methodology and empirically validate it. For example, showing that the theoretical burden of the O(n^2) scaling does not apply in practice.
- I like Appendix Section A2. While more speculative (and therefore probably part of the appendix), I like that the authors try to assess the real-world implications of this pairing scheme (which at first might seem limited because this is not how we currently conduct peer review).

Weaknesses:
- Since we have no access to the ground truth when implementing this system in real time, it would be helpful if the authors could offer either theoretical bounds or heuristics on how many pairs should be evaluated before we converge to some high-quality ranking.
- The authors acknowledge this limitation already partially in Appendix Section A1, but still the endogeneity effect of citations (being accepted to the conference may bump citations) and using human reviewer decisions (e.g., accept vs reject) as a validation criterion seems to contradict the statement that current peer review is failing us as we try to replicate its outcomes. Maybe the authors should clarify the stance that the mechanism how we conduct peer review could be improved, but the outcomes themselves we're trying to replicate.

---

> ### Author Rebuttal · Authors · 2025-07-30
>
> Thank you very much for your helpful feedback and support for the paper! We have carefully updated the paper to incorporate your suggestions.
>
> > Since we have no access to the ground truth when implementing this system in real time, it would be helpful if the authors could offer either theoretical bounds or heuristics on how many pairs should be evaluated before we converge to some high-quality ranking.
>
> Thank you for the suggestion. Following your guidance, We have added a theoretical bound (Theorem 4 in [1]) to clarify how many pairwise comparisons are needed to recover a high-quality global ranking. Specifically, it shows that if $ m > 12n \log n $ pairs are sampled uniformly at random, then with high probability, the estimate $ \hat{\theta} $ will satisfy
>
> $$
> \Vert \hat{\theta} - \theta^* \Vert = O\left(n \sqrt{\frac{\log n}{m}}\right)
> $$
>
> This provides a theoretical guideline for how many comparisons are required to achieve reliable ranking performance, even when the true scores are not accessible during real-time setting.
>
> > Maybe the authors should clarify the stance that the mechanism how we conduct peer review could be improved, but the outcomes themselves we're trying to replicate.
>
> Thank you for the suggestion. We have added discussion clarifying our stance on improving the peer review process versus replicating its outcomes as follows:“ Ideally, a strong alternative review mechanism should correlate with—but not fully replicate—the outcomes of the current system, offering complementary perspectives and surfacing different strengths. As we show, our proposed pairwise comparison mechanism achieves this balance, aligning with current human judgments while introducing useful differentiation. Crucially, our goal is not to mimic existing outcomes, but to design mechanisms that add value. ”
>
> > Are the differences in the novel research topics statistically significant? There seems to be a large overlap between the distributions.
>
> Thank you for your question. Although there is a large overlap between the distributions, the differences are statistically significant (p < 0.05) for ICLR 2023 Notable-top-5%, ICLR 2024 Oral, and EMNLP 2023 Main Track. For CoRL 2023 Oral and NeurIPS 2023 Oral, we did not observe statistically significant differences, which may be due to smaller sample sizes. We have added a clarification in the corresponding experiments section of the paper.
>
> > Do the authors have any suggestions on how to incorporate responses to the meta-reviews? The back and forth (e.g., clarifications, additional experiments) between reviewers and authors is an important part of the review process which is omitted using this approach.
>
> Thank you for raising the important question of how to incorporate responses to the meta-reviews. In our study, we focus on the **decision-making layer** of peer review—specifically, mechanisms that can rank submissions and support acceptance decisions. While our current framework does not explicitly incorporate such back-and-forth exchanges, we agree that doing so could enrich the review process by capturing meaningful clarifications and author revisions.
>
> One natural extension would be to introduce a reranking phase after the initial comparisons. Specifically, once authors have responded to meta-reviews, all submissions could be updated and then re-included in a second round of pairwise polling. To better capture the effect of these interactions, both original and revised versions could be included in the reranking phase. Comparing their relative positions allows us to study how author responses influence reviewer preferences.
>
> We have added the following paragraph to clarify that our framework is intended to broaden, not replace, the collaborative dynamics that characterize high-quality peer review: “ Moreover, we believe the strength of human-led peer review lies not only in final decisions, but also in the human interactions it enables—feedback, discussion, and iterative refinement—that help authors strengthen their work. Through these interactions, reviewers and authors also engage in a shared evaluative practice that reinforces norms, builds trust, and helps individuals identify with the scholarly community [2, 3]. Our intention is not to automate or replace this process, but to broaden how we understand and support scholarly evaluation.”
>
> **References**
>
> [1] Sahand Negahban, Sewoong Oh, and Devavrat Shah. (2017). Rank Centrality: Ranking from Pairwise Comparisons. Oper. Res. 65, 1 (January-February 2017), 266–287. https://doi.org/10.1287/opre.2016.1534
>
> [2] Zuckerman, H., & Merton, R. K. (1971). Patterns of evaluation in science: Institutionalisation, structure and functions of the referee system. Minerva, 66-100.
>
> [3] Lamont, M. (2009). How professors think: Inside the curious world of academic judgment. Harvard University Press.

---

### Official Review · Reviewer_vHzj · 2025-07-11

**Clarity:** 2
**Significance:** 2
**Originality:** 3
**Rating:** 4
**Confidence:** 3

**Summary:**

This paper proposes a novel redesign of LLM-based peer review by shifting from conventional per-paper scoring to pairwise comparison of manuscripts. Instead of assigning absolute scores, LLM agents compare two papers at a time and vote for the better one. These results are aggregated using the Bradley–Terry model to produce a global ranking of all submissions. Experiments on major ML conference papers (e.g., ICLR, NeurIPS, EMNLP) demonstrate that this method better identifies high-impact papers (measured by citation count) than traditional rating-based GPT review systems. However, the method also exhibits biases: a preference for less novel topics and overrepresentation of papers from elite institutions.

**Questions:**

see weakness

**Ethical Concerns:**

["NO or VERY MINOR ethics concerns only"]

**Final Justification:**

I  maintain my original overall score 4 ( boardline accept).

**Quality:**

2

**Strengths And Weaknesses:**

Strengths:
1. The motivation is clear and make sense. The authors argue convincingly that the current peer review system is a historically contingent artifact, and LLMs allow rethinking the entire structure.
2. The paper is well-written.
3. The idea is simple but novel in review paradigm: The paper proposes a genuinely different peer review workflow using pairwise comparisons, diverging from the current mainstream of LLM-based review replication.

Weaknesses:
1. Bias Amplification: The approach tends to favor papers from top institutions and penalize novelty, raising serious fairness and diversity concerns.

2. Limited Model Diversity: Experiments rely solely on GPT-4o mini. It's unclear whether the findings generalize across other LLMs.

3. Citation as Sole Impact Metric: The paper assumes that citation count is an accurate proxy for quality, which can be problematic due to citation inflation, visibility bias, and Matthew effects.

---

> ### Author Rebuttal · Authors · 2025-07-30
>
> We thank Reviewer vHzj for their positive comments and helpful feedback on our work.
>
> > Bias Amplification: The approach tends to favor papers from top institutions and penalize novelty, raising serious fairness and diversity concerns.
>
> Thank you for the comment. As noted in the discussion, while our LLM-based peer review approach shows promise, it also raises concerns around fairness and diversity, such as institutional concentration and reduced topic novelty. We see these not as definitive flaws, but as opportunities to inform more equitable and inclusive review practices. Our system is intended to augment—not replace—the human review process, and we hope this analysis inspires further research on mitigation strategies and alternative designs aligned with academic values.
>
> > Limited Model Diversity: Experiments rely solely on GPT-4o mini. It's unclear whether the findings generalize across other LLMs.
>
> Thank you for the suggestion. We have conducted experiments to test two additional models other than GPT-4o mini: Gemini 2.0 Flash and Claude-3-haiku-20240307. We scaled the number of comparisons to over $10^6$ and found that, in both cases, our pairwise comparison framework performs significantly better than the GPT rating system baseline. Specifically, papers selected by Gemini 2.0  Flash receive an average of 18.3 citations (vs 11.4), while those chosen by Claude‑3‑Haiku average 16.8 (vs 11.4). These results suggest that the advantage of our approach is robust across different LLMs.
>
> > Citation as Sole Impact Metric: The paper assumes that citation count is an accurate proxy for quality, which can be problematic due to citation inflation, visibility bias, and Matthew effects.
>
> Thank you for raising this important point. We acknowledge that citation counts, while widely used, are an imperfect reflection of research quality. As described in Appendix A.1, they are affected by numerous factors beyond quality, such as the visibility of research communities, prevailing research trends, and established reputations.
>
>
> To further verify the robustness of our proposed method, we conducted experiments using alternative measures. Specifically, we measured the Spearman correlation between our system’s output and human review scores. In both ICLR 2023 and ICLR 2024, the BT scores produced by our mechanism show moderate correlation with the average human ratings—27% and 24%, respectively. This indicates that our system aligns with signals in the current review process, while not fully replicating its outcomes. We believe this complementary nature is essential for identifying potential improvements to existing peer review systems.
>
> Moreover, we  added explicit clarification to emphasize that these metrics are used as a practical means of evaluation: “ It is important to note that while we use proxy metrics—such as future citation counts—to evaluate our proposed mechanism, this represents an exploratory and pragmatic step rather than an ideal. The gold standard would be direct human evaluation of how well a mechanism supports the selection and improvement of high-quality research, but such evaluations are costly and difficult to scale [1]. Proxy metrics offer a tractable, though imperfect, way to approximate long-term impact and anticipate how different mechanisms might perform. Ideally, a strong alternative review mechanism should correlate with—but not fully replicate—the outcomes of the current system, offering complementary perspectives and surfacing different strengths. As we show, our proposed pairwise comparison mechanism achieves this balance, aligning with current human judgments while introducing useful differentiation. Crucially, our goal is not to mimic existing outcomes, but to design mechanisms that add value.”
>
> [1] Si, Chenglei, Diyi Yang, and Tatsunori Hashimoto. "Can LLMs Generate Novel Research Ideas? A Large-Scale Human Study with 100+ NLP Researchers." The Thirteenth International Conference on Learning Representations.

---

### Decision · Program_Chairs · 2025-09-17

**Decision:**

Accept (poster)

**Comment:**

This paper proposes using LLM agents to perform pairwise comparisons of manuscripts rather than individual scoring. This new approach to peer review demonstrates improved identification of high-impact papers across multiple conferences compared to rating-based methods, while the author(s) analyzed concerning biases including reduced novelty and institutional imbalance. Despite reviewer concerns about limited novelty and evaluation metrics, the thorough experimental validation across different LLMs and the comprehensive examination of system limitations are solid contributions. Neverthess, the author(s) should address concerns raised by the reviewers in the final version.